

# A survey of HK, HPt, and RR domains and their organization in two-component systems and phosphorelay proteins of organisms with fully sequenced genomes

Baldiri Salvado[1], Ester Vilaprinyo[1,2], Albert Sorribas[1] and Rui Alves[1]

[1] Departament de Ciencies Mèdiques Bàsiques, Universitat de Lleida, Lleida, Catalonya, Spain
[2] IRBLleida, Lleida, Catalonya, Spain

## ABSTRACT

Two Component Systems and Phosphorelays (TCS/PR) are environmental signal transduction cascades in prokaryotes and, less frequently, in eukaryotes. The internal domain organization of proteins and the topology of TCS/PR cascades play an important role in shaping the responses of the circuits. It is thus important to maintain updated censuses of TCS/PR proteins in order to identify the various topologies used by nature and enable a systematic study of the dynamics associated with those topologies. To create such a census, we analyzed the proteomes of 7,609 organisms from all domains of life with fully sequenced and annotated genomes. To begin, we survey each proteome searching for proteins containing domains that are associated with internal signal transmission within TCS/PR: Histidine Kinase (HK), Response Regulator (RR) and Histidine Phosphotranfer (HPt) domains, and analyze how these domains are arranged in the individual proteins. Then, we find all types of operon organization and calculate how much more likely are proteins that contain TCS/PR domains to be coded by neighboring genes than one would expect from the genome background of each organism. Finally, we analyze if the fusion of domains into single TCS/PR proteins is more frequently observed than one might expect from the background of each proteome. We find 50 alternative ways in which the HK, HPt, and RR domains are observed to organize into single proteins. In prokaryotes, TCS/PR coding genes tend to be clustered in operons. 90% of all proteins identified in this study contain just one of the three domains, while 8% of the remaining proteins combine one copy of an HK, a RR, and/or an HPt domain. In eukaryotes, 25% of all TCS/PR proteins have more than one domain. These results might have implications for how signals are internally transmitted within TCS/PR cascades. These implications could explain the selection of the various designs in alternative circumstances.

Corresponding author
Rui Alves, ralves@cmb.udl.es

## INTRODUCTION

Historically, Two Component Systems and Phosphorelays (TCS/PR) have been considered primary environmental signal transduction cascades in prokaryotes (*Wolanin, Thomason & Stock, 2002*; *Cheung & Hendrickson, 2010*). In TCS/PR, environmental signals regulate the autophosphorylation state of a sensor histidine kinase. In TCS this sensor transfers its phosphate to a response regulator, which will in turn directly regulate the relevant cellular responses to the signal. The sensor and response regulator may be two independent proteins. They may also be the same protein, containing independent domains that are responsible for each of the two functions. In PR, additional phosphotransfer steps may happen before the phosphate reaches the response regulator protein(s) that directly controls cellular responses (Fig. 1). PR are considered to be a main form of signal transduction in bacteria (*Parkinson, 1993*; *Hoch & Silhavy, 1995*). They are less frequently present in eukaryotes and absent in animals (*Chang et al., 1993*; *Maeda, Wurgler-Murphy & Saito, 1994*; *Appleby, Parkinson & Bourret, 1996*; *Thomason & Kay, 2000*).

The mechanism of signal sensing in the various types of TCS/PR have been studied with great detail and is reviewed elsewhere (*Inouye & Dutta, 2003*; *Simon, Crane & Crane, 2007*; *Gross & Beier, 2012*). Extensive and insightful reviews have also been published about the topology (pattern of molecular interactions between the proteins in the cascade), crosstalk and signal transmission in TCS/PR (*Oka, Sakai & Iwakoshi, 2002*; *Majdalani & Gottesman, 2005*; *Bekker, Teixeira de Mattos & Hellingwerf, 2006*; *Laub & Goulian, 2007*; *Szurmant, White & Hoch, 2007*; *Ortiz de Orué Lucana & Groves, 2009*; *Krell et al., 2010*; *Buelow & Raivio, 2010*; *Szurmant & Hoch, 2010*; *Hazelbauer & Lai, 2010*; *Casino, Rubio & Marina, 2010*; *Porter, Wadhams & Armitage, 2011*; *Schaller, Shiu & Armitage, 2011*; *Kobir et al., 2011*; *Seshasayee & Luscombe, 2011*; *Gross & Beier, 2012*; *Jung et al., 2012*; *Podgornaia & Laub, 2013*; *Fassler & West, 2013*; *Mascher, 2014*), as well as about the domain structure and evolution of the proteins involved in the cascades (*Inouye & Dutta, 2003*; *Catlett, Yoder & Turgeon, 2003*; *Galperin & Nikolskaya, 2007*; *Cock & Whitworth, 2007*; *Jenal & Galperin, 2009*; *Whitworth & Cock, 2009*; *Kim et al., 2010*; *Wuichet, Cantwell & Zhulin, 2010*; *Cheung & Hendrickson, 2010*; *Galperin, 2010*; *Perry, Koteva & Wright, 2011*; *Seshasayee & Luscombe, 2011*; *Capra & Laub, 2012*; *Sheng et al., 2012*; *Attwood, 2013*; *Ortet et al., 2015*).

There are several protein types and domains that nature uses in TCS/PR cascades. For example, CHEW adapter proteins permit transmitting information about nutrient gradients to the TCS system that regulates bacterial response to those gradients (*Jones & Armitage, 2015*). In another example, the PII protein regulates the activity of the TCS that responds to nitrogen depletion in the environment (*Huergo et al., 2012*). There are other cases where external proteins bind proteins from a TCS/PR cascade and modulate their stability (*Salvado et al., 2012*). These protein types are used in TCS/PR with specific biological functions and are not common to all TCS/PR cascades.

Nevertheless, there are four types of protein domains that are common to all TCS/PR cascades. Sensor domains, with wide sequence variability, are responsible for capturing the environmental changes and adjusting the activity of the cascade (*Cheung & Hendrickson, 2010*; *Hazelbauer & Lai, 2010*). Irrespective of protein domain organization, signal

**Figure 1 Two component systems.** (A) Prototypical two component system with one phosphotransfer step between HK and RR. (B) 3-step phosphorelay, with four protein domains involved in the signal transduction process and 3 phosphotransfer steps.

transmission within a TCS/PR circuit is done using histidine kinase (HK) domains, response regulator (RR) domains, and/or histidine phosphotransfer (HPt) domains. These last three domains are responsible for internal signal transmission (IST) within the cascade and represent the focus of the current work. Because they are common to all TCS/PR cascades, the results from our study are generally applicable and do not depend on the specific environmental signal or biological response mediated by the cascades.

Several examples demonstrate that the dynamic range and signal-response curve that a given cascade might exhibit is closely related to the interactions between the various proteins and to the organization of IST domains within each cascade protein (*Alves & Savageau, 2003*; *Igoshin et al., 2004*; *Igoshin, Alves & Savageau, 2008*; *Eswaramoorthy et al., 2010*; *Ray, Tabor & Igoshin, 2011*; *Salvado et al., 2012*; *Narula et al., 2012*). For example, circuits where each IST domain is in an independent protein are more likely to participate in cross-talk and branching is more likely to occur in the signal transduction process (*Catlett, Yoder & Turgeon, 2003*; *Seshasayee & Luscombe, 2011*). In addition, noise propagates differently in a cascade of independent IST domain proteins than in a cascade where IST domains are found within the same protein (*Swain, 2004*) (Fig. 2). Also, TCS where phosphatases are involved in dephosphorylating the response regulator protein may show hysteretic behavior. In contrast, TCS where the sensor protein works both as phosphodonor and phosphatase for the response regulator may only exhibit graded responses to changes in the signal (*Igoshin, Alves & Savageau, 2008*).

These and other examples (*Alves & Savageau, 2003*; *Süel et al., 2006*; *Kuchina et al., 2011a*; *Kuchina et al., 2011b*) show that connectivity of the TCS/PR circuits and domain organization of the proteins play an important role in shaping the responses of the cascades to their cognate signals. It is thus important to maintain censuses of TCS/PR proteins in order to identify the various network topologies used by nature and enable a systematic study of the internal signal transduction dynamics associated with those topologies. Information about such topologies can be retrieved for a detailed analysis from several databases (*Ulrich & Zhulin, 2010*; *Finn et al., 2014*).

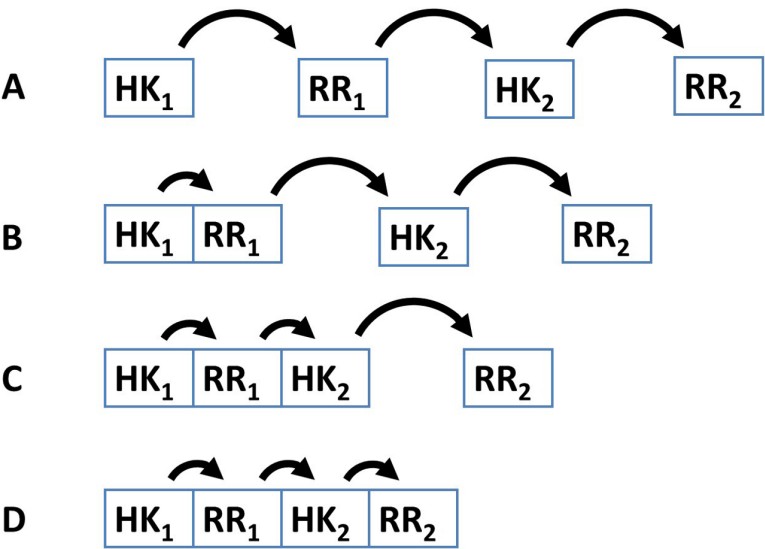

**Figure 2** **Four different patterns of covalent linkage between the protein domains involved in phosphorelays.** (A) A four protein phosphorelay. (B) A phosphorelay with and hybrid kinase at the beginning of the cascade. (C) A two protein phosphorelay where the first two phosphotransfer steps between domains contained in a single protein. (D) A one protein phosphorelay, where all phosphotransfer steps take place between domains of a single protein.

While MIST2 (*Ulrich & Zhulin, 2010*) contains information about less than 3,000 genomes, Pfam contains a few hundred sequences divided among the HK, RR, and HPt domain families involved in TCS/PR cascades. Currently, at the NIH there are over 10,000 fully sequenced and annotated genomes that are freely accessible to the public. Because of this, obtaining a more up to date census of the TCS/PR in these genomes is an important task that we set out to do. We analyzed the TCS/PR proteins of 7,609 organisms from all domains of life with fully sequenced and annotated genomes. We focus on the IST domain families HK, RR, and HPt of TCS/PR cascades. First, we survey the number of TCS/PR domains in each organism and how these domains are arranged into individual proteins. Then, we find all different type of operon organizations and analyze how much more likely are proteins that contain TCS/PR domains to be coded by neighboring genes than one would expect from the genome background. Finally, we analyze how the percentage IST domain fusion within TCS/PR proteins changes among all analyzed genomes.

Our census finds that there are 50 alternative ways in which the HK, HPt, and RR domains are observed to organize into single proteins. 90% of all proteins identified in this study contain just one RR or HK domain, while 8% of the remaining proteins combine one copy of a HK, a RR, and/or a HPt domain. We also find that more than 25% of all TCS/PR eukaryotic proteins have more than one domain. Our results are consistent with previous works and identify TCS/PR proteins in all non-animal phyla. Overall, our results set the stage for a systematic study to compare the internal dynamic behavior of signal transduction associated with each circuit topology in TCS/PR cascades.

## MATERIAL AND METHODS

### Identification of proteins containing TCS/PR domains

The fully annotated proteomes of 9,961 organisms were downloaded from NCBI's genome database (January 2014 version). 2,352 of these proteomes were eliminated because they belonged to phages, virus, satellite DNA sequences, or organisms whose taxonomic classification was still not fully resolved. The remaining 7,609 proteomes belonging to 35 phyla from Bacteria, 6 phyla from Archaea and 11 eukaryotic phyla (Table S1) were further analyzed in search for proteins containing domains of types HK, RR and HPt. These domains are associated with IST in all TCS/PR cascades. Other protein domains (such as the CHEW adaptor domain or the P2 protein from NRI/NRII, among many others) were not included in the analysis because they are specific of certain TCS/PR cascades. The sensor domain of TCS/PR cascade proteins was also not included due to its sequence variability. Thus, the results from our study are general for all TCS/PR cascades.

We used PROSITE (http://prosite.expasy.org) to obtain a set of well curated sequences that can be used as a seed to identify TCS/PR proteins in the relevant proteomes. We downloaded a multiple alignment of all relevant ortholog sequences for each protein domain (HK—PS50109 PROSITE Domain, RR—PS50110 PROSITE Domain and HPt—PS50894 PROSITE Domain) from PROSITE. We then used these three multiple alignments as a set of query sequences for two independent searches. One was done using HMMER (*Johnson, Eddy & Portugaly, 2010*). For each multiple alignment downloaded from PROSITE, we built a profile HMM using hmmbuild, and performed the search of the profile HMM against all proteomes selected from the NCBI database using jackhmmer. The second search was done in parallel using PSI-BLAST (*Altschul et al., 1990*) and the three multiple alignments downloaded from PROSITE as a query. HMMER finds homologues that are more distantly related than those found by BLAST.

We simultaneously use BLAST and HMMer because they have different sensitivities in detecting sequence similarities. BLAST generates a higher number of false negatives, while HMMer generates a higher number of false positives. By using both and filtering the results, we hope to obtain a more precise picture of the conserved domains. In each search we queried the 7,609 proteomes in order to identify proteins with domains that are homologous to those used as queries.

In addition, the consensus sequence was calculated for each domain (HK, RR and HPt) independently. Using an in-house PERL script, the most common residue in each position was identified for each of the three multiple alignments. This residue was taken as the consensus value for that position in the corresponding protein domain. Subsequently the three consensus sequences were used to search each proteome using PSI-BLAST (*Altschul et al., 1990*). In all three searches, the hits selected were the ones with an e-value lower than $10^{-6}$ and with a domain coverage of at least 80%.

After performing these three searches, a PERL script was also used to perform a fourth text-mining search and identify the proteins that were annotated in each proteome as being histidine kinases, sensory kinases, hybrid kinases, response regulators or histidine phosphotransferases.

The results of the four searches were merged into a non-redundant set. A total amount of 469,421 proteins containing HK, HPt and/or RR domains were identified. This set was curated in the following way:

(1) First we manually looked at the annotation of the proteins to identify functions that are not involved in TCS/PR cascades (e.g., serine kinase).
(2) Then, we build a PERL script that automatically eliminates proteins annotated with those functions from the list.
(3) We finish by automatically comparing the proteins in the list and the number of proteins containing terms related to TCS/PR cascades.
(4) We repeat steps 1–3 until the number of proteins in the list and the number of proteins containing only terms related to TCS/PR cascades are the same.

In this way, we semi manually identified 36,169 proteins that were annotated as being something other than a TCS/PR protein. These proteins were eliminated. Frequent protein types found in the discarded set of proteins are serine/threonine kinases and several types of regulatory transcription factors.

The remaining 433,255 proteins were then reanalyzed and an additional set of 17,727 proteins were found to be annotated as being hypothetical or partial proteins. For each of the three domains, the set of 17,727 hypothetical and partial proteins were aligned using Clustal X in order to identify the conserved histidine motif in the HK and HPt domains, and the conserved aspartate residue in RR domains. Those sequences without a conserved histidine or aspartate residue were eliminated from the data, leaving a grand total of 415,525 annotated proteins and 17,724 partial/hypothetical proteins containing HK, RR and/or HPt domains.

A PERL script was developed to filter the curated data sets and determine both, the domain composition of each protein and, when they belonged to the same organism, the relative position of their corresponding genes with respect to each other in the genome.

Once we had identified all proteins containing HK, RR or HPt domains, and the relative genomic position of their corresponding genes, we looked for all type of operons of TCS/PR coding genes that occur in the organisms with fully sequenced genomes. For this purpose, we performed a search of all genes coding HK, RR or HPt protein domains that are located in consecutive positions on prokaryotic genomes. We assumed that they constitute a transcription unit, although this may introduce a small error, as consecutive operons coding for independent TCS/PR exceptionally exist. In our search, we allow the presence of a gap in the operon, that is, a gene which does not encode any HK, RR or HPt domain, because this could be a gene with regulatory functions in the operon.

The statistical treatment of data was carried out independently with and without taking into account the hypothetical and partial proteins found. Both results are qualitatively the same. In the Results section of this paper we give the results from the analysis of the set of proteins without the hypothetical and partial proteins. The results of the whole set of proteins, including hypothetical and partial proteins, are given as Supplemental

## Numerical and statistical data analysis

To estimate how the clustering of the various TCS and PR proteins in a genome differed from what one would expect by chance in the context of that genome, we took the following approach. First, we calculated how frequently one would expect proteins containing TCS/PR domains to be coded by neighboring genes in a genome if the order of genes was fully random, given the total number of proteins in that genome, and the number of proteins involved in TCS/PR cascades. The expected neighboring frequencies under this assumption can be computed by Eqs. (1)–(6). In these $F(P1 \leftrightarrow P2)$ represents the expected frequency of the neighboring events in a genome for genes coding proteins of types $P1$ and $P2$, $n_{RR}$ represents the number of proteins containing one RR domain in the proteome, $n_{HK}$ represents the number of proteins containing one HK domain in the proteome, and $P$ represents the total number of proteins annotated to the proteome.

$$F(\text{HK} \leftrightarrow \text{RR}) = \frac{n_{RR}}{P-1} + \frac{n_{RR}}{P-2}. \tag{1}$$

Equation (1) represents the probability that a gene localized in position $j$ of the genome is located next to a gene coding for a protein that contains an RR domain, either in positions $j - 1$ or $j + 1$, if gene order is random in a genome. The first term of the sum represents the probability of the presence of an RR gene in one of the two possible locations irrespective of its presence also in the other genome location, and the second term is the probability of the presence of the RR gene in one of the two genome locations when it is not found in the other one. We note that we are not calculating the probability of having a consecutive gene pair containing HK and RR domains. Rather, for any genomic position $j$, we ask what the probability of its neighboring a gene containing an RR domain is. Equation (1) gives a good estimation of this random probability, given that the total number of protein coding genes is tens to hundreds of times larger than the number of IST domain coding genes, and assuming that position $j$ represents neither the first nor the last genomic position. This expected RR neighboring frequency will be compared with the actual fraction of HK genes that are found next to RR genes in order to study their genomic distribution.

$$F(\text{HK} \leftrightarrow \text{RR} \leftrightarrow \text{HK}_2) = 6 \times \frac{n_{HK}}{P-1} \times \frac{n_{RR}}{P-2}. \tag{2}$$

Equation (2) computes the probability of finding an RR gene and a second HK gene in the genomic neighborhood of a given HK gene. Because these three consecutive genes can be sorted in 6 different ways, we must multiply by 6 the probability of an individual neighboring event. Again, note that we assume having an HK domain containing gene, and ask what the probability of its neighboring genes containing additional HK and RR domains is.

$$F(\text{HK} \leftrightarrow \text{RR} \leftrightarrow \text{HK}_2 \leftrightarrow \text{RR}_2) = 12 \times \frac{n_{RR}}{P-1} \times \frac{n_{HK}-1}{P-2} \times \frac{n_{RR}-1}{P-3}. \tag{3}$$

Similarly, in Eq. (3) we compute the probability that, considering that we have found a gene containing an HK domain in a given place in the genome, we also find in consecutive genomic positions around that HK gene location another HK gene and two RR genes, if gene organization is random. These four genes can be sorted in 24 different ways, but we don't differentiate between the two RR genes and therefore there are only 12 possible spatial arrangements of these series of four genes.

$$F(\text{HKRR} \leftrightarrow \text{HK} \leftrightarrow \text{RR}) = 6 \times \frac{n_{\text{HK}}}{P-1} \times \frac{n_{\text{RR}}}{P-2}. \tag{4}$$

In Eq. (4), the probability of the event is computed in exactly the same way as in Eq. (2).

$$F(\text{HKRRHPt} \leftrightarrow \text{RR}) = \frac{n_{\text{RR}}}{P-1} + \frac{n_{\text{RR}}}{P-2} \tag{5}$$

$$F(\text{HKRRHK} \leftrightarrow \text{RR}) = \frac{n_{\text{RR}}}{P-1} + \frac{n_{\text{RR}}}{P-2}. \tag{6}$$

Equations (5) and (6) compute the probability of finding an RR gene placed in the genome next to an HKRRHPt or an HKRRHK gene respectively, exactly in the same way as described above for Eq. (1).

Once these expected frequencies were computed using Eqs. (1)–(6), we calculated the odds ratios of the observed neighboring events with respect to the expected neighboring event. All numerical and statistical calculations were done using Mathematica (*Wolfram, 1999*).

## Statistical models

To analyze the relationship between the number of TCS/PR gene fusion events and the proteome size, we built a linear model that would better fit our data for % of fused HK (RR, HPt) domains vs. total number of HK (respectively, RR, HPt). We also built linear models of total number of IST domains in an organism vs. total number of proteins in the proteome and phylogeny (prokaryote, eukaryote). In other words, we fit the data to Eq. (7):

$$\text{Number of IST domains} = \alpha_1 \left( \text{Total number of proteins in proteome} \right)$$
$$+ \alpha_2 \, \text{Phylogeny} + \varepsilon. \tag{7}$$

In Eq. (7), the variable phylogeny can assume two values. If the organism is a prokaryote, the variable has value 1; otherwise it has value 2. An ANOVA analysis was used to determine whether the coefficients for each control variable of the linear model are significantly different from zero. If so, this implies that the variable is relevant in explaining the variation observed in the dependent variable.

When fitting the data to the linear models we also calculated the $R^2$ and adjusted $R^2$ of the models. $R^2$ shows how well terms (data points) fit a curve or line; adjusted $R^2$ also indicates how well terms fit a curve or line, but adjusts for the number of terms in a model.

## RESULTS

### Survey of proteomes containing proteins with domains involved in internal signal transduction (IST) in TCS/PR cascades

#### Bacteria

Table 1 summarizes the full set of results for bacteria. Proteins with HK and RR domains are present in the proteome of 100% of the species analyzed from the following bacterial phyla: Aquificae, Chlorobi, Verrucomicrobia, Chloroflexi, Cyanobacteria, Deferribacteres, Deinococcus-Thermus, Dictyoglomi, Acidobacteria, Nitrospirae, Planctomycetes, Epsilonproteobacteria, Spirochaetes, Thermodesulfobacteria, and Thermotogae. In contrast, proteins containing HK and/or RR domains were not identified in a small percentage of species in the following phyla: Actinobacteria—0.63% (4 out of 635 species surveyed), Bacteroidetes—9.36% (22 out of 235 species), Firmicutes—0.68% (14 out of 2066), Fusobacteria—5.26% (2 out of 38), Alphaproteobacteria—3.55% (16 out of 451), Betaproteobacteria—1.64% (6 out of 366), Deltaproteobacteria—1.22% (1 out of 82), Epsilonproteobacteria—0.24% (1 out of 410), Gammaproteobacteria—1.83% (41 out of 2246), Synergistetes—9.09 % (1 out of 11). Interestingly, no proteins containing HK or RR domains were identified in most Tenericutes species. Only 18 out of the 111 surveyed Tenericutes species have proteins with HK and RR domains.

The percentage of species in each phylum with proteins containing HPt domains is lower than the percentage of species with HKs and RRs, and ranges from less than 10% (Chlamydiae, Tenericutes) to more than 90% (Deferribacteres, Acidobacteria, Nitrospirae, Planctomycetes, Deltaproteobacteria, Epsilonproteobacteria, Gammaproteobacteria, Spirochaetes, Thermodesulfobacteria, and Thermotogae). It should be noted that HPt domains are also used by proteins that import PTS sugars (*Clore & Venditti, 2013*), which means that not all HPt domains we found are involved in PR or TCS signal transduction.

#### Archaea

Proteins with HK and RR domains were identified in the proteome of 154 out of 179 Euryarchaeota species, 9 of the 11 Taumarchaeota species and only 2 out of 51 Crenarchaeota species surveyed. Proteins with HPt domains were identified in the proteome of 115 Euryarchaeota species and in 7 of the 11 Taumarchaeota species surveyed. No proteins containing HK, RR, or HPt domains were identified in Nanoarchaeota, Nanohaloarcheota, and Korarchaeota (Table 1).

#### Eukaryotes

HK and RR domains were identified in the proteomes of 20 in 35 fungi species. 32 fungi species contain proteins where the HPt-domain was identified. HK, HPt, and RR domains were identified in the proteomes of the 2 eudicot species surveyed, but only HK and HPt, and not RR domains, were identified in *Oryza sativa*.

There are only two surveyed protist phyla that contain proteins with IST domains. These phyla are Euglenozoa and Amoeboflagellates. We analyzed five Euglenozoa species. Out of these, only *Leishmania donovani* and *Leishmania major* contain proteins with HK and RR IST domains. These domains are always found in separate proteins. Interestingly,

**Table 1  Percentage of species in each phylum with TCS/PR proteins.**

| Domain | Phylum | Abbreviaton | No. of species surveyed | % of species with HK and RR domains | % of species with HPt domains |
|---|---|---|---|---|---|
| Bacteria | Actinobacteria | At | 635 | 99.37 | 14.49 |
| Bacteria | Aquificae | Aq | 13 | 100.00 | 76.92 |
| Bacteria | Armatimonadetes | Ar | 1 | 100.00 | 100.00 |
| Bacteria | Bacteroidetes | Ba | 235 | 89.79 | 49.79 |
| Bacteria | Chlorobi | Cb | 14 | 100.00 | 71.43 |
| Bacteria | Caldiserica | Cd | 1 | 100.00 | 0.00 |
| Bacteria | Chlamydiae | Cm | 108 | 98.15 | 1.85 |
| Bacteria | Lentisphaerae | L | 1 | 100.00 | 100.00 |
| Bacteria | Verrucomicrobia | V | 10 | 100.00 | 80.00 |
| Bacteria | Chloroflexi | Cf | 23 | 100.00 | 65.21 |
| Bacteria | Chrysiogenetes | Cr | 1 | 100.00 | 100.00 |
| Bacteria | Cyanobacteria | Cy | 118 | 100.00 | 75.42 |
| Bacteria | Deferribacteres | Df | 4 | 100.00 | 100.00 |
| Bacteria | Deinococcus-Thermus | Dt | 20 | 100.00 | 35.00 |
| Bacteria | Dictyoglomi | Dc | 2 | 100.00 | 0.00 |
| Bacteria | Elusimicrobia | El | 1 | 100.00 | 0.00 |
| Bacteria | Acidobacteria | Ac | 9 | 100.00 | 100.00 |
| Bacteria | Fibrobacteres | Fb | 1 | 100.00 | 100.00 |
| Bacteria | Firmicutes | Fi | 2,066 | 99.42 | 37.80 |
| Bacteria | Fusobacteria | Fu | 38 | 94.74 | 28.95 |
| Bacteria | Gemmatimonadetes | Ge | 1 | 100.00 | 100.00 |
| Bacteria | Nitrospinae | Ni | 1 | 100.00 | 100.00 |
| Bacteria | Nitrospirae | Nt | 4 | 100.00 | 100.00 |
| Bacteria | Planctomycetes | Pl | 20 | 100.00 | 100.00 |
| Bacteria | Alphaproteobacteria | A | 451 | 96.67 | 58.31 |
| Bacteria | Betaproteobacteria | B | 366 | 98.36 | 59.56 |
| Bacteria | Deltaproteobacteria | D | 82 | 98.78 | 98.78 |
| Bacteria | Epsilonproteobacteria | E | 410 | 100.00 | 98.54 |
| Bacteria | Gammaproteobacteria | G | 2,246 | 98.31 | 95.46 |
| Bacteria | Zetaproteobacteria | Z | 1 | 100.00 | 100.00 |
| Bacteria | Spirochaetes | S | 274 | 100.00 | 99.64 |
| Bacteria | Synergistetes | Sy | 11 | 90.91 | 63.64 |
| Bacteria | Tenericutes | T | 111 | 15.32 | 7.21 |
| Bacteria | Thermodesulfobacteria | Th | 2 | 100.00 | 100.00 |
| Bacteria | Thermotogae | Tt | 17 | 100.00 | 100.00 |
| Archaea | Crenarchaeota | C | 51 | 3.92 | 3.92 |
| Archaea | Euryarchaeota | Eu | 179 | 86.03 | 64.25 |
| Archaea | Korarchaeota | K | 1 | 0.00 | 0.00 |
| Archaea | Thaumarchaeota | Ta | 11 | 81.82 | 63.64 |
| Archaea | Nanoarchaeota | N | 1 | 0.00 | 0.00 |
| Archaea | Nanohaloarchaeota | Nh | 1 | 0.00 | 0.00 |
| Eukarya | Alveolates | Av | 5 | 0.00 | 20.00 |

Table 1 (*continued*)

| Domain | Phylum | Abbreviaton | No. of species surveyed | % of species with HK and RR domains | % of species with HPt domains |
|---|---|---|---|---|---|
| Eukarya | Amoeboflagellates | Am | 1 | 100.00 | 100.00 |
| Eukarya | Euglenozoa | Eg | 5 | 40.00 | 0.00 |
| Eukarya | Microsporidians | Mi | 2 | 50.00 | 0.00 |
| Eukarya | Ascomycetes | As | 31 | 54.84 | 96.77 |
| Eukarya | Basidiomycetes | Bs | 2 | 100.00 | 100.00 |
| Eukarya | Eudicots | Ed | 2 | 100.00 | 100.00 |
| Eukarya | Monocots | M | 1 | 0.00 | 100.00 |
| Eukarya | Nematodes | – | 1 | 0.00 | 0.00 |
| Eukarya | Arthropods | – | 7 | 0.00 | 0.00 |
| Eukarya | Chordates | – | 10 | 0.00 | 0.00 |

only one RR domain containing protein was identified in each of the two species. Surprisingly, only HK domains were identified in proteins from *Leishmania_infantum* and *Trypanosoma_brucei*. No IST domains were identified in *Leishmania_braziliensis*. In Amoeboflagellates, only *Dictyostelium discoideum* has proteins containing IST domains in its proteome. The HK domain was only identified in hybrid HKRR, HKRR1RR2, or HKRR1HK2RR2 proteins. In contrast, RR domains also appear in proteins where no other IST domains are identified.

No HK, RR, or HPt domains were found in animal proteomes in the context of TCS/PR cascades.

## Percentage of proteins with HK, RR or HPt domains in the surveyed proteomes

For simplicity, hereafter we shall refer to proteins containing IST domains typical from TCS/PR cascades as TCS/PR proteins. On average, between 1 and 2% of a prokaryotic proteome is composed of TCS/PR proteins (mean = 1.37%). In contrast, when an eukaryotic proteome contains TCS/PR proteins, they account for between 0.05% and 0.2% of the entire proteome (mean = 0.11%). In bacteria, Deltaproteobacteria is the group with the highest average percentage of TCS/PR proteins (Fig. 3). In contrast Tenericutes and Chlamydiae almost tie with the lowest average percentage of TCS/PR proteins (Fig. 3).

It has been observed in previous analyses that the number of proteins containing IST domains associated with TCS/PR cascades increases almost quadratically with the number of total proteins in a proteome (*Ulrich, Koonin & Zhulin, 2005*; *Galperin, Higdon & Kolker, 2010*). We further wanted to assess if this dependency is significantly different between eukaryotes and prokaryotes. To do so we fit the data to the linear model described by Eq. (7). An ANOVA analysis shows that phylogeny is important in explaining the variation in total number of IST domains found in a proteome ($p < 10^{-25}$). Because of this we divided the dataset in prokaryotes and eukaryotes, and fit each dataset to the linear model shown in Fig. 4. We find that the fraction of variability in number of IST domains explained by proteome size in eukaryotes doubles that of prokaryotes. This suggests that the number of IST domains could evolve differently in prokaryotic and eukaryotic organisms.

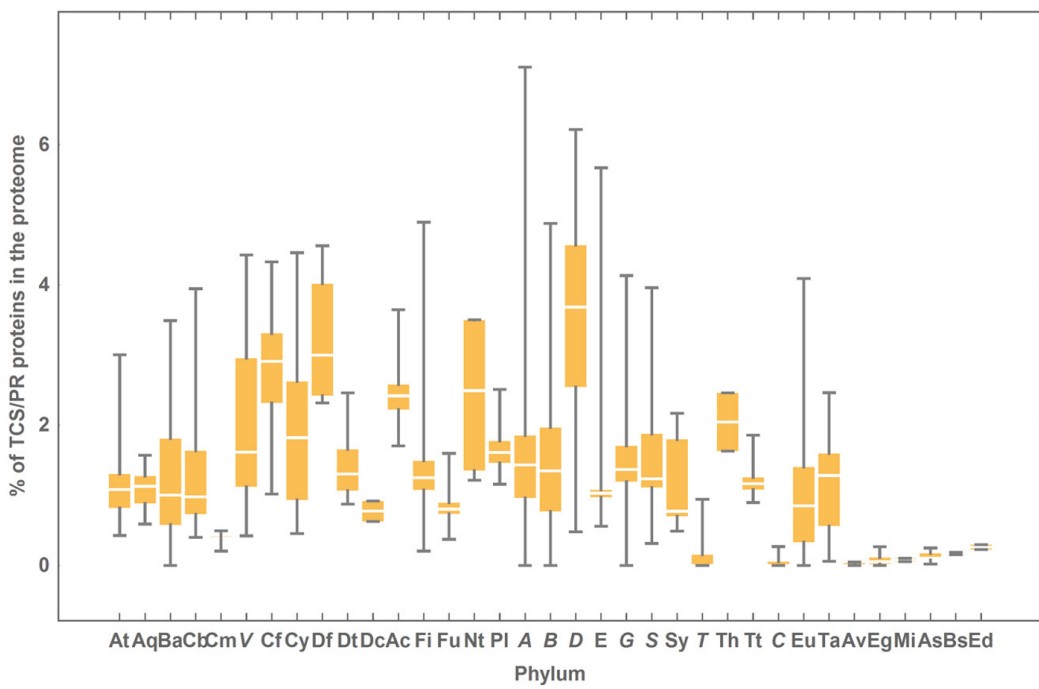

**Figure 3 Percentage of TCS/PR proteins in the proteome per phylum.** The colored box represents the range of percentage values comprised between the 25% and the 75% quantiles, and the edges of the vertical bar denote the upper and lower percentage values for each phylum. Phylum abbreviations are given in Table 1. Phyla with only one species surveyed are not represented in the figure. Their percentage of TCS/PR proteins per phylum are: Aq (1.30), Ge (3.15), Fb (0.81), L (0.68), Cr (3.73), El (0.78), Ar (0.93), Z (2.47), O (4.95), Ni (1.97), K (0), N (0), Nh (0), Am (0.17) and M (0.04). We have found only 2 TCS/PR proteins in Av (5 sp): 1 HPt in *T. annulata* and 1 HK in *T. parva*.

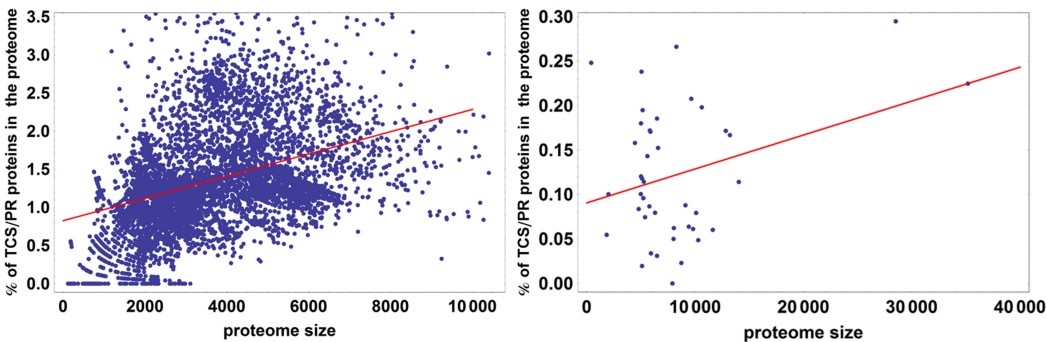

**Figure 4 Percentage of TCS/PR proteins in the proteome versus total number of proteins in the proteome.** $R^2$ is 0.21 for prokaryotes and 0.49 for eukaryotes. This means that proteome size explains 21% of the variation in the percentage of TCS/PR in prokaryotes and 49% in eukaryotes.

## Survey of TCS/PR protein types

We find fifty unique types of TCS/PR proteins, when it comes to IST domain organization within a single polypeptide chain. These unique types of TCS/PR proteins are shown in Table 2, sorted by abundance. In that table, the protein identifier describes the type of IST domain (HK, HPt, or RR) and the number describes how many domains of a given IST

**Table 2 Types of TCS/PR proteins found in the 7,609 surveyed species.** The protein identifier describes the type (HK, HPt, or RR) and number of TCS/PR domains fused in each protein.

| Protein type | Total number of proteins found | Percentage of proteomes with this type of protein | Number of species with this type of protein | Average number of proteins/organism |
|---|---|---|---|---|
| RR | 219,436 | 97,07 | 7,386 | 29.71 |
| HK | 151,849 | 95,98 | 7,303 | 20.79 |
| HKRR | 18,383 | 48,57 | 3,696 | 4.97 |
| HKRRHPt | 9,097 | 40,85 | 3,108 | 2.93 |
| HKHPt | 5,506 | 41,99 | 3,195 | 1.72 |
| HPt | 3,534 | 28,05 | 2,134 | 1.66 |
| $RR_1RR_2$ | 2,034 | 17,60 | 1,339 | 1.52 |
| $HKRR_1RR_2$ | 2,017 | 13,59 | 1,034 | 1.95 |
| $HKRR_1HPtRR_2$ | 982 | 8.12 | 618 | 1.59 |
| $HK_1RR_1RR_2RR_3$ | 580 | 6.58 | 501 | 1.16 |
| $HK_1HK_2$ | 450 | 4.07 | 310 | 1.45 |
| $HK_1RRHK_2$ | 392 | 3.30 | 251 | 1.56 |
| RRHPt | 312 | 3.47 | 264 | 1.18 |
| $HKRRHPt_1HPt_2HPt_3$ | 141 | 1.85 | 141 | 1.00 |
| $RR_1RR_2HPt$ | 130 | 1.45 | 110 | 1.18 |
| $HKRRHPt_1HPt_2HPt_3HPt_4$ | 108 | 1.42 | 108 | 1.00 |
| $HK_1RR_1HK_2RR_2$ | 90 | 0.79 | 60 | 1.50 |
| $RR_1RR_2RR_3HPt$ | 72 | 0.51 | 39 | 1.85 |
| $HKRRHPt_1HPt_2HPt_3HPt_4HPt_5$ | 61 | 0.80 | 61 | 1.00 |
| $HKRRHPt_1HPt_2$ | 58 | 0.72 | 55 | 1.05 |
| $HK_1HK_2RRHPt$ | 39 | 0.50 | 38 | 1.03 |
| $HK_1HK_2HPt$ | 39 | 0.50 | 38 | 1.03 |
| $HKHPt_1HPt_2$ | 36 | 0.46 | 35 | 1.03 |
| $RR_1RR_2RR_3$ | 34 | 0.32 | 24 | 1.42 |
| $HKRR_1RR_2RR_3HPt$ | 33 | 0.37 | 28 | 1.18 |
| $HPt_1HPt_2$ | 21 | 0.20 | 15 | 1.40 |
| $HKHPt_1HPt_2HPt_3$ | 16 | 0.20 | 15 | 1.07 |
| $HK_1HK_2RR_1RR_2RR_3$ | 9 | 0.12 | 9 | 1.00 |
| $HK_1HK_2HK_3$ | 9 | 0.04 | 3 | 3.00 |
| $HKRR_1RR_2RR_3RR_4RR_5HPt$ | 7 | 0.09 | 7 | 1.00 |
| $HKRRHPt_1HPt_2HPt_3HPt_4HPt_5HPt_6HPt_7$ | 7 | 0.09 | 7 | 1.00 |
| $HKRR_1RR_2RR_3RR_4$ | 6 | 0.08 | 6 | 1.00 |
| $HK_1HK_2HK_3HK_4RR_1RR_2$ | 6 | 0.08 | 6 | 1.00 |
| $HK_1HK_2RRHPt_1HPt_2$ | 5 | 0.07 | 5 | 1.00 |
| $HKRR_1RR_2RR_3RR_4HPt$ | 5 | 0.07 | 5 | 1.00 |
| $RR_1RR_2RR_3RR_4$ | 2 | 0.03 | 2 | 1.00 |
| $HK_1HK_2RR_1RR_2HPt_1HPt_2$ | 2 | 0.03 | 2 | 1.00 |
| $HK_1HK_2RR_1RR_2RR_3RR_4$ | 2 | 0.03 | 2 | 1.00 |
| $HK_1HK_2HK_3HK_4$ | 2 | 0.03 | 2 | 1.00 |
| $HK_1HK_2HPt_1HPt_2$ | 2 | 0.03 | 2 | 1.00 |

Table 2 (*continued*)

| Protein type | Total number of proteins found | Percentage of proteomes with this type of protein | Number of species with this type of protein | Average number of proteins/organism |
|---|---|---|---|---|
| $HKRR_1RR_2HPt_1HPt_2$ | 2 | 0.03 | 2 | 1.00 |
| $HK_1HK_2HK_3RR$ | 1 | 0.01 | 1 | 1.00 |
| $HPt_1HPt_2HPt_3$ | 1 | 0.01 | 1 | 1.00 |
| $HK_1HK_2RRHPt_1HPt_2HPt_3$ | 1 | 0.01 | 1 | 1.00 |
| $HKRR_1RR_2RR_3HPt_1HPt_2HPt_3$ | 1 | 0.01 | 1 | 1.00 |
| $HPt_1HPt_2HPt_3HPt_4$ | 1 | 0.01 | 1 | 1.00 |
| $HKRR_1RR_2HPt_1HPt_2HPt_3$ | 1 | 0.01 | 1 | 1.00 |
| $HK_1HK_2RR_1RR_2HPt$ | 1 | 0.01 | 1 | 1.00 |
| $HK_1HK_2RR_1RR_2RR_3RR_4RR_5RR_6HPt$ | 1 | 0.01 | 1 | 1.00 |
| $HKRRHPt_1HPt_2HPt_3HPt_4HPt_5HPt_6$ | 1 | 0.01 | 1 | 1.00 |

type are found in each protein. Hereafter we shall refer to proteins containing only one HK IST domain as HK protein type, proteins containing one HK domain and one RR domain as HKRR protein type, and so on and so forth.

Overall, all phyla where IST domains associated with TCS/PR cascades were identified have RR and HK protein types, with the exception of Monocots, which lack RR domains. HKRR protein type (also known as hybrid HK) is present in all phyla where TCS/PR proteins were identified, except in Aquificae, Tenericutes, Chlamydiae, and Crenarchaeota (Table S2). Together, HK, RR, and HKRR proteins represent 94% of all TCS/PR proteins that were identified.

In prokaryotes, RR or HK protein types are the most abundant. Together, they represent more than 90% of all TCS/PR proteins found in the genomes of many organisms (Table S2). HKRR represent the third most abundant type of TCS/PR protein, oscillating between less than 1% (Firmicutes) and more than 10% (Cyanobacteria) of all TCS/PR proteins (Table S2). The remaining protein types ($HPt, HKRRHPt, HK_1RRHK_2, HKRR_1HPtRR_2, HK_1RR_1HK_2RR_2, \ldots$) range from less than 1% to 5% of all TCS/PR proteins identified in a phylum. Of these less abundant protein types, the three-domain HKRRHPt protein is more abundant than $HK_1RRHK_2$. The HPt domain is more frequently found in combination with other IST TCS/PR protein domains than alone in a protein, with the exception of Firmicutes, Tenericutes, Actinobacteria, Bacteroidetes and Spirochaetes. We also observe that $HKRR_1HPtRR_2$ is more abundant than $HK_1RR_1HK_2RR_2$ (Table S2).

The relative abundances of proteins containing IST domains associated with TCS/PR cascades in eukaryotes are different from those of prokaryotes. In broad terms, HK and RR protein types tend to make for a smaller fraction of TCS/PR proteins in eukaryotes than in prokaryotes, while the opposite is observed for HKRR proteins. Another clear distinction between prokaryotes and eukaryotes refers to HPt-containing proteins: HPt protein type represents more than 10% of all TCS/PR proteins in eukaryotes. In prokaryotes, except in Tenericutes, HPt proteins typically account for less than 1% of TCS/PR proteins. Moreover, no HKRRHPt or $HKRR_1HPtRR_2$ protein types were found in eukaryotes (Table S2).

Among protists, Euglenozoa proteomes contain mostly HK protein type, although HKRR type is the most abundant in *D. discoideum* (Amoeboflagellate). There are cases of inactive HK domains that have lost their histidine. When identified, these proteins were eliminated from the analysis as described in Methods. However, there is always the possibility that some such proteins have passed our filters. To control for that possibility we created a multiple alignment of the Euglenozoa HK proteins. We found that the HK domains contained the conserved histidine motif that is needed for HK signal transduction. Hence, these proteins could be active HK proteins. Furthermore, if we lower our e-value for cut-off to $10^{-4}$, many of these proteins will also be flagged as containing RR domains with conserved aspartate residues, suggesting that such proteins could be HKRR types with a high degree of sequence divergence from other HKRR proteins we identified. Thus, the HK proteins in this clade might either be hybrid HKs or be active in a context that does not involve a TCS/PR cascade. TCS/PR proteins are almost absent in Alveolates. In the fungi phyla (Table 1), HKRR is the most abundant protein type in Basidiomycetes, making up for almost 50% of total TCS/PR proteins. In contrast, RR, HK and HPt protein types are relatively more abundant than HKRR protein type in Ascomycetes. A remarkable result in fungi is the relative abundance of $HK_1RRHK_2$ and $HK_1RR_1HK_2RR_2$, which are much more frequent in eukaryotes (above 10%) than in prokaryotes. In plants, RR is the most abundant protein type in Eudicots, making for about 60% of all TCS/PR proteins.

## Distribution of genes coding for TCS/PR protein types in the genomes

Previous surveys found that many of the TCS/PR proteins are mostly organized in operons and/or regulons in prokaryotes (*Alm, Huang & Arkin, 2006*; *Williams & Whitworth, 2010*; *Galperin, 2010*; *Galperin, Higdon & Kolker, 2010*). Consistent with this, we find that between 60% and 90% of genes containing HK domains are neighbors to genes containing RR domains. Exact percentages depend on the phylum, but below 20% of the total prokaryotic HK coding genes are orphan, that is, they are not neighboring any other gene coding for a protein that contains at least one IST domain. We also have found some clusters of genes coding HK, RR or HPt domains in eukaryotes, but all of them are a succession of genes with identical domain composition. Although the existence of operons has been reported in the eukaryote *C. elegans* (*Blumenthal, Davis & Garrido-Lecca, 2015*), the gene clusters identified in our search have independent promoters.

Altogether, we found 530 different types of gene clusters coding for TCS/PR proteins. We now briefly describe these results, shown in Table S9.

## Neighborhood analysis for HK and RR protein types

In most prokaryotes neighboring genes coding for HK and RR protein types are between 50 and 100 times more frequent than one might expect by chance alone. In some species, this frequency is even higher (Fig. S1 and Table S3). Several phyla have a small percentage of species containing only orphan HK and RR protein types in their genomes (20 out of 2066 species in Firmicutes, 2 out of 635 in Actinobacteria, 6 out of 235 in Bacteroidetes, 11 out of 2,246 in Gammaproteobacteria, 48 out of 451 in Alphaproteobacteria, 7 out of 366

in Betaproteobacteria, 4 out of 108 in Chlamydiae, 3 out of 118 in Cyanobacteria and 9 out of 179 in Euryarchaeota). Most of these species have a number of TCS/PR proteins below the average of their phylum.

### Neighborhood analysis for HK-RR-HK2

Approximately 20% of all prokaryotic species have HK-RR-HK2 consecutive genes in their genomes at least 10 (and sometimes 50) times more frequently than one might expect by chance alone. Conversely, the frequency of this gene neighborhood organization is what one would expect by chance alone in the remaining 80% prokaryotic species (Table S4).

### Neighborhood analysis for HK-RR-HK2-RR2

In most prokaryotic phyla, between 10% and 60% of species have *HK-RR-HK2-RR2* genes at least 100 times more frequently than one would expect by chance alone (Table S5).

### Neighborhood analysis for HKRR-HK2-RR2

In the majority of prokaryotic species, genes coding for proteins of type HKRR have no neighboring genes coding for proteins of types HK or RR. Nevertheless, in more than 20% of the species of some prokaryotic phyla, such as Proteobacteria or Spirochaetes, genes coding for HKRR-protein type are neighbors to genes coding for HK or RR protein type with a frequency more than 100 times higher than expected by chance alone (Table S6).

### Neighborhood analysis for HKRRHPt next to RR2

In most of the prokaryotic species where HKRRHPt protein types are present, the observed frequency of HKRRHPt-RR genetic neighborhoods is between 10 and 50 times more frequent than one would expect by chance alone (Table S7).

### Neighborhood analysis for HK1RRHK2 next to RR2

In prokaryotes, $HK_1RRHK_2$ is a scarce protein, present only in a few species (Table 2). If present, it is located in the genome next to a RR protein type on average 31% of the times (Table 3). In Gammaproteobacteria, $HK_1RRHK_2$ is present only in 28 out of 2246 species surveyed, and in 9 of these 28 species, the observed frequency of $HK_1RRHK_2$ genes placed in the chromosome next to RR genes is more than 100 times higher than the random expected frequency (Table S8).

## Gene fusion of TCS/PR proteins

### Gene fusion events

The number of gene fusion events observed in a genome is expected to be proportional to genome size, in a model for neutral evolution of protein domain fusion (*Durrens, Nikolski & Sherman, 2008*; *Whitworth & Cock, 2009*). Thus, if gene fusion events in the case of HK and RR are random one would expect that the linear model that would best fit the data for % of fused HK (RR, HPt) domains vs. total number of HK (respectively, RR, HPt) domains has slope zero. In contrast, if these events are favored, the slope of that model should be positive, and if the events are disfavored, that slope should be negative.

We analyze fusion events of IST domains associated with TCS/PR cascades in the individual phyla by creating a linear model of percentage of fused HK (or RR) domains

**Table 3 Total number of HKRRHPt and HKRRHK proteins found in prokaryotic phyla.** Phyla in bold are from the bacterial domain. Italicized phyla are from the archaeal domain.

| Phylum | Number of HKRRHPt/$HK_1RRHK_2$ proteins found | Number of HKRRHPt/$HK_1RRHK_2$ genes with a neighboring RR gene | % of HKRRHPt/$HK_1RRHK_2$ genes with a neighboring RR gene |
|---|---|---|---|
| Actinobacteria | 12/4 | 9/1 | 75.00/25.00 |
| Aquificae | 0/0 | 0/0 | —/— |
| Armatimonadetes | 0/0 | 0/0 | —/— |
| Bacteroidetes | 107/9 | 62/4 | 57.94/44.44 |
| Chlorobi | 4/0 | 0/0 | 0.00/— |
| Caldiserica | 0/0 | 0/0 | —/— |
| Chlamydiae | 2/0 | 1/0 | 50.00/— |
| Lentisphaerae | 1/0 | 0/0 | 0.00/— |
| Verrucomicrobia | 12/2 | 9/1 | 75.00/50.00 |
| Chloroflexi | 16/0 | 8/0 | 50.00/— |
| Chrysiogenetes | 1/0 | 0/0 | 0.00/— |
| Cyanobacteria | 193/28 | 41/9 | 21.24/32.14 |
| Deferribacteres | 9/0 | 7/0 | 77.78/— |
| Deinococcus-Thermus | 0/4 | 0/1 | —/25.00 |
| Dictyoglomi | 0/0 | 0/0 | —/— |
| Elusimicrobia | 0/0 | 0/0 | —/— |
| Acidobacteria | 1/5 | 1/2 | 100.00/40.00 |
| Fibrobacteres | 0/0 | 0/0 | —/— |
| Firmicutes | 65/97 | 44/69 | 67.69/71.13 |
| Fusobacteria | 2/0 | 2/0 | 100.00/— |
| Gemmatimonadetes | 3/0 | 3/0 | 100.00/— |
| Nitrospinae | 0/0 | 0/0 | —/— |
| Nitrospirae | 4/0 | 3/0 | 75.00/— |
| Planctomycetes | 40/0 | 18/0 | 45.00/— |
| Alphaproteobacteria | 337/10 | 233/5 | 69.14/50.00 |
| Betaproteobacteria | 364/9 | 274/4 | 75.27/44.44 |
| Deltaproteobacteria | 208/29 | 131/1 | 62.98/3.45 |
| Epsilonproteobacteria | 399/0 | 389/0 | 97.49/— |
| Gammaproteobacteria | 7239/28 | 3336/15 | 46.08/53.57 |
| Zetaproteobacteria | 2/0 | 1/0 | 50.00/— |
| Spirochaetes | 53/147 | 16/3 | 30.19/2.04 |
| Synergistetes | 6/0 | 6/0 | 100.00/— |
| Tenericutes | 0/0 | 0/0 | —/— |
| Thermodesulfobacteria | 2/0 | 1/0 | 50.00/— |
| Thermotogae | 6/0 | 5/0 | 83.33/— |
| *Crenarchaeota* | 0/0 | 0/0 | —/— |
| *Euryarchaeota* | 9/1 | 3/0 | 33.33/0.00 |
| *Thaumarchaeota* | 0/0 | 0/0 | —/— |
| **Total** | **9097/373** | **4603/115** | **50.60/30.83** |

as a function of the total number of HK (or RR) domains in the genome and calculate the likelihood that the slope is different from zero. The results are shown in Table 4. We find that the percentage of fused HK (or RR) domains increases with the number of HK (or RR) domains in the genomes. This is consistent with a positive selection for fused HKRR proteins.

## DISCUSSION

### Scope, caveats, and limitations of our analysis

In this work we analyze the distribution and prevalence of different types of TCS/PR proteins in 7609 organisms belonging to 52 phyla. These proteins are responsible for sensing and adequately regulating the cellular responses to environmental cues. To date, this is the largest survey of TCS/PR proteins we are aware of. We confirm that these proteins are predominantly prokaryotic, although they are also present in many eukaryotic phyla. However, functional TCS/PR cascades appear to be absent in animals. This is also consistent with previous findings (*Attwood, 2013*).

An important feature in this study is that we include all organisms with fully sequenced and annotated genomes in our analysis. For example, on the order of one thousand *Escherichia coli* strains are included in our analysis. This would clearly bias any deletion/duplication or horizontal gene transfer study of TCS/PR proteins that one might make in the full dataset. However, considering all strains and subspecies in our analysis is fundamental for identifying extremely low-frequency unique IST domain and operon organization types.

### Identifying unique types of IST domain organization in TCS/PR cascades

The main goal of this analysis is to identify the unique types of organization for IST domains in proteins of TCS/PR cascades. In addition we also perform a less thorough identification of operon organization for TCS/PR proteins. This study was independently made in two ways: first, we eliminate all proteins annotated as hypothetical or partial. Subsequently we include such proteins in the analysis. The results for the analysis that include the hypothetical and partial proteins can be found as a ZIP supplementary file (Appendix S1). Results are qualitatively similar in both cases, and the raw sequences in FASTA format can be downloaded from http://web.udl.es/usuaris/pg193845/Salvadoretal.html.

Our analysis identifies 50 unique types of TCS/PR proteins, when it comes to intra protein IST domain organization. The most frequent types of proteins with fused IST domains are the hybrid histidine kinases, a design with one HK and one RR protein domains fused in a single protein. This organization has been observed in most of the eukaryotic PRs that have been well characterized genetically and biochemically (for example the Sln1p-Ypd1p-Ssk1p pathway in *S. cerevisiae* (*Maeda, Wurgler-Murphy & Saito, 1994*) or the ETR1 system in *A. thaliana* (*Chang et al., 1993*)). It is also present in some prokaryotic systems (for example, the RcsC/YojN/RcsB pathway, involved in the regulation of capsular polysaccharide synthesis in *E. coli* (*Takeda et al., 2001*), and the Lux pathway regulating bioluminescence in *V. harveyi* (*Freeman & Bassler, 1999*)). Another

**Table 4 Percentage of RR and HK domains in hybrid proteins as a function of the total number of HK and RR proteins in the genome.** Phyla in bold are from the bacterial domain. Italicized phyla are from the archaeal domain. Other phyla are from the eukaryotic domain.

| Phylum | RR | SK |
|---|---|---|
| Gammaproteobacteria | $6.97 + 0.2x^{**}$ | $14 + 0.31x^{**}$ |
| Betaproteobacteria | $1.90 + 0.22x^{**}$ | $4.6 + 0.37x^{**}$ |
| Epsilonproteobacteria | $15.3 - 0.06x^{***}$ | $0.17 + 0.36x^{*}$ |
| Deltaproteobacteria | $17.3 + 0.1x^{*}$ | $31.9 + 0.06x^{***}$ |
| Alphaproteobacteria | $4.1 + 0.29x^{**}$ | $3.8 + 0.4x^{**}$ |
| Firmicutes | $-0.6 + 0.1x^{**}$ | $1.7 + 0.09x^{*}$ |
| Tenericutes | – | – |
| Actinobacteria | $-4.5 + 0.32x^{**}$ | $-0.38 + 0.2x^{*}$ |
| Chlamydiae | – | – |
| Spirochaetes | $5 + 0.47x^{*}$ | $27.2 + 0.07x^{***}$ |
| Acidobacteria | $-5.7 + 0.26x$ | $-16 + 0.53x^{*}$ |
| Bacteroidetes | $30.7 + 0.05x^{***}$ | $32 + 0.09x^{***}$ |
| Fusobacteria | $-5.8 + x$ | $-7.1 + 1.5x$ |
| Verrumicrobia | $6.7 + 0.3x^{***}$ | $6.6 + 0.4x^{***}$ |
| Planctomycetes | $32.8 - 0.1x^{***}$ | $49.8 - 0.21x^{***}$ |
| Synergistetes | – | – |
| Cyanobacteria | $2.2 + 0.3x^{**}$ | $5.8 + 0.4x^{**}$ |
| Green sulfur bacteria | $31.6 + 0.7x^{***}$ | $33.5 + 0.5x^{***}$ |
| Green non-sulfur bacteria | $5.2 + 0.2x$ | $8.9 + 0.2x$ |
| Deinococcus-Thermus | $-1.2 + 0.2x$ | $-1.6 + 0.2x$ |
| Euryarchaeota | $6.9 + 0.6x^{*}$ | $15.4 + 0.1x^{***}$ |
| Crenarchaeota | – | – |
| Nanoarchaeota | – | – |
| Korarchaeota | – | – |
| Oomycetes | – | – |
| Diatoms | – | – |
| Parabasilids | – | – |
| Diplomonads | – | – |
| Euglenozoa | – | – |
| Alveolates | – | $9.5 + 0.4x^{***}$ |
| Amoeboflagellates | – | – |
| Choanoflagellates | – | – |
| Microsporideans | – | – |
| Basidiomycetes | $25.7 + 3.7x$ | $88 + 1.1x^{***}$ |
| Ascomycetes | $37.4 + 2.5x^{**}$ | $92.4 - 0.07x^{***}$ |
| Red algae | – | – |
| Green algae | $29.2 + x^{***}$ | $114.7 - 9x^{***}$ |
| Mosses | – | – |
| Monocots | $12.3 + 0.2x^{***}$ | $32.9 + 1.9x^{***}$ |
| Eudicots | $17.4 + 0.1x^{***}$ | $71 - 0.5x^{***}$ |

**Notes.**

[*] $p$-value $< 10^{-3}$

[**] $p$-value $< 10^{-8}$

[***] Non-significant ($p$-value $> 0.1$)

relatively frequent type of IST domain organization is when one HK, one HPt, and one RR domain are found within a single protein. Such proteins are called unorthodox histidine kinase or tripartite HK. Some examples of systems with this design are: BvgS-BvgA (*Uhl & Miller, 1996*), EvgS/EvgA (*Bock & Gross, 2002*), ArcB/ArcA (*Georgellis, Lynch & Lin, 1997*), TorS/TorR (*Bordi et al., 2003*), BarA/UvrY (*Sahu et al., 2003*), TodS/TodT (*Silva-Jiménez, Ramos & Krell, 2012*) and GacS/GacA (*Sahu et al., 2003*).

We also identify 530 unique types of possible operons in prokaryotes and some eukaryotes, such as ascomycetes and eudicots (Table S9). This variety will be used in subsequent works to infer naturally occurring variations in the pattern of regulatory interactions between the proteins involved in TCS/PR networks. For example, if we find a gene cluster formed by one HK and two RR coding genes, we can infer that the signaling pathway has a branching point in which the HK phosphorylates both RR. This alternative circuitry is important because it has been proved that network architecture affects network dynamics and can define the operational limits of the system in a way that is independent of the specific biological processes being regulated (*Alves & Savageau, 2003*; *Igoshin, Alves & Savageau, 2008*; *Cağatay et al., 2009*; *Tiwari et al., 2010*; *Salvado et al., 2012*).

We have no way of identifying TCS/PR cascades at the regulon level using only sequence data. Many examples for this type of organization exist, such as the Kin-SpoO pathway (*Burbulys, Trach & Hoch, 1991*).

Why do we focus only on the IST domains of TCS/PR cascades, rather than also including also other protein domain that are involved in TCS/PR signal transduction? By focusing on these domains and their organization, our results set the stage for an analysis of general dynamics organization principles in the internal transmission of signals within TCS/PR cascades. The organization of IST domains, either within a protein or within an operon, plays an important role in determining the dynamics of the signal transmission in a cascade (*Alves & Savageau, 2003*; *Igoshin, Alves & Savageau, 2008*; *Ray & Igoshin, 2010*; *Narula et al., 2012*). Hence, that organization is likely to be the subject of natural selection. Had we included other types of domains, we would be also analyzing aspects of the input and output of the cascades that are case specific and not general to all cascades of a given type.

## Some physiological, phylogenetic and evolutionary considerations

In prokaryotes, approximately 90% of all PR proteins have only one HK domain or one RR domain (Table 2 and Table S2), and most of the genes encoding these proteins are located in the chromosome next to other PR/TCS genes, forming operons. In contrast to this, in eukaryotes proteins of types HK and RR are less common, and genes encoding these proteins are never located next to other TCS/PR genes in the species surveyed. On the other hand, in eukaryotes there is a higher fraction of TCS/PR proteins containing a combination of the HK and RR domains (the HPt domain was not found in these eukaryotic multi domain TCS/PR proteins), such as HKRR, $HK_1RRHK_2$ and $HK_1RR_1HK_2RR_2$. This implies that TCS/PR signal transduction in eukaryotes is in principle less prone to cross-talk and noise, as the signal is internally transmitted within the same peptide chain (*Tiwari et al., 2010*; *Tiwari & Igoshin, 2012*).

Our analysis confirms the *ad hoc* observation that coordinated expression of IST domains and/or TCS/PR proteins involved in the same cascade is frequent. We also quantify how much more frequent this coordinated expression is with respect to what one would expect by chance alone. Although this is not unexpected (*Price, Arkin & Alm, 2006*; *Alm, Huang & Arkin, 2006*; *Ray & Igoshin, 2012*; *Tiwari & Igoshin, 2012*), to our knowledge, such quantification had not been done before on such a large dataset.

This suggests that alternative IST regimes might be favored by evolution in prokaryotic or eukaryotic TCS/PR cascades. This can be inferred from the fact that the three types of gene expression coordination (regulon, operon, or gene fusion) imply different characteristics when it comes to internal signal transmission within the cascade. In general, fused genes will have a lower level of noise in signal transduction, followed by genes coded in the same operon, and with genes coded in the same regulon permitting the highest level of noise to enter the signal transduction process (*Ray & Igoshin, 2012*).

Why is this so? TCS/PR proteins whose expression is coordinated either at the regulon or operon levels are potentially translated in different amounts. RR proteins typically are orders of magnitude more abundant than HK proteins (*Igoshin, Alves & Savageau, 2008*). This leads to a type of signal transduction where amplification of the signal can be high, as many RR molecules can be modified by a single HK protein. In contrast, in hybrid kinases where the HK and RR domains are fused in the same protein, the ratio of HK/RR domains is one to one. This means that each HK domain will likely only phosphorylate one RR domain. Moreover, independent HK protein types might also be leakier, phosphorylating non-cognate RRs. Similarly, independent RR protein types can be more prone to phosphorylation by non-cognate sources. Such non-cognate phosphorylation events are physically harder to achieve in HKRR protein types. Thus, proteins with fused TCS/PR domains represent a design that will on average transduce signals with smaller amplification, but higher fidelity than TCS/PR cascades composed only of proteins with individual TCS/PR domains.

Taking these considerations into account, one might think that maximization of internal signal amplification is likely to be an important selective pressure for the evolution of TCS/PR cascades in prokaryotes, while fidelity of internal signal transmission appears to be a more important selective pressure for the evolution of TCS/PR cascades in eukaryotes. These two functional requirements for IST in TCS/PR cascades are generic and independent of more specific pressures, such as the type of signal they transduce, whether the organism is uni- or multi-cellular, or other similar considerations (*Alm, Huang & Arkin, 2006*; *Laub & Goulian, 2007*; *Williams & Whitworth, 2010*; *Galperin, Higdon & Kolker, 2010*; *Capra & Laub, 2012*; *Podgornaia & Laub, 2013*). If and why amplification and fidelity of internal signal transmission are indeed shaping the general organization of TCS/PR cascades is a matter to be investigated further in the future. This will be done in a forthcoming study by creating mathematical models for the TCS/PR cascade architectures identified in this study and comparing the dynamic behavior of each of the alternatives.

### Funding

This work was partially funded by Grant BFU2010-17704 from the Spanish MINECO and from small grants CMB and TR255 from University of Lleida to RA. The funders had no role in study design, data collection and analysis, decision to publish, or preparation of the manuscript.

### Grant Disclosures

The following grant information was disclosed by the authors:
Spanish MINECO: BFU2010-17704.
University of Lleida: CMB and TR255.

### Competing Interests

The authors declare there are no competing interests.

### Author Contributions

- Baldiri Salvado and Rui Alves conceived and designed the experiments, performed the experiments, analyzed the data, contributed reagents/materials/analysis tools, wrote the paper, prepared figures and/or tables, reviewed drafts of the paper.
- Ester Vilaprinyo analyzed the data, contributed reagents/materials/analysis tools, wrote the paper, prepared figures and/or tables, reviewed drafts of the paper.
- Albert Sorribas analyzed the data, wrote the paper, reviewed drafts of the paper.

### Supplemental Information

Supplemental information for this article can be found online at http://dx.doi.org/10.7717/peerj.1183#supplemental-information.

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
