# Peer review of "A survey of HK, HPt, and RR domains and their organization in two-component systems and phosphorelay proteins of organisms with fully sequenced genomes"

_PeerJ, doi:10.7717/peerj.1183_

## Round 0.1 · original submission · Major Revisions

Based on the reports of the referees, I regret to inform you that the paper is unsuitable for publication in its present form.

The main concerns are related to statistical data analyses, the domain identification pipeline, and the overlooking of important phylogenetic features such as gain/losses/duplication events of genes within the phyla. It also requires a careful proofreading by a native speaker, and the lack of a discussion in terms of Biological relevance is also an important issue. In its current form, it reads like a cataloging effort exercise.

However, if the paper were substantially rewritten taking into account the aforementioned observations, and the referees' comments, especially those regarding the statistical analyses and the domain identification procedures, it may become acceptable for publication.

Reviewer 1 ·

Basic reporting

The submission does adhere to PeerJ policies except for analyses of non-labelled and hypothetical proteins. This data is the most interesting because it is less known. An Open Access journal demands in my opinion access to all data with no 'data not shown' statements (line 375).
Clear English.
I found at least 5 papers that are not cited in the ms, are relevant to this work and consider some comments or at least statement that such work has been done [Biochem Soc Trans. 2013 Aug;41(4):1023-8, Biosci Biotechnol Biochem. 2010;74(4):716-20, Curr Opin Microbiol. 2010 Apr;13(2):219-25, Adv Exp Med Biol. 2012;751:121-37, Nucleic Acids Res. 2015 Jan;43(Database issue):D536-41]. All these articles are closely related to work presented in here. If not cited I would expect an explanation why not.
Structure is OK.
Relevant figures.

Experimental design

The submission is quite original, however similar works have already been done, but not to that extent, according to my knowledge.
The research question is clearly stated, however the title suggests a broader analysis:
- the analysis does not take into account the local events, at the level of phyla or lower. The title 'Distribution...' means also local bursts/duplications/losses of TCSs. With the unprecedented number of data that task seems obvious and consistent with the title of the manuscript.
- I also do not see any relation to function. This subject is of course completely aside of the main area of the manuscript but I believe that authors didn't want solely to catalogue the TCSs but also to find a biological sense in it. At least somewhere in the Discussion I would expect any relation with biological properties of the findings.
Methods OK.
Ethics OK.

Validity of the findings

Data is statistically sound. My suggestions are as follows:
- why use HMMER and BLAST when there are jackHMMER and PSI-BLAST?
- line 129: sequences were aligned using what programme?
- lines 140-145: how can you eliminate 8% out of half a million hits manually? How did you make it? Using annotations (which can be false...)? Please describe in more detail the "semiautomatic" way.
It would be great if the detailed results of the study were deposited in any OA databases, like MIST or P2CS.
Conclusions are OK but mostly look more like a Summary. Maybe more conclusions can be inferred from your data? :)

Additional questions/comments:
- line 133: numbered citation of the BLAST paper
- line 146: 'were' instead of 'where'
- line 220-222: proteomes instead of proteome.
- lines 270-290: I didn't get if you finally incorporated the 'inactive' HK domains, didn't you?
- Figure 3: in the description there should be full names of the phyla.
- Figure 4: R2 is not explained.

Additional comments

Your work is great but needs two major elements:
- incorporation of your data into existing databases,
- any relation of your work with biological function/ecological niches to answer the question 'Why is it so different in between species?' In my opinion it is the major drawback of this interesting work.

Reviewer 2 ·

Basic reporting

The authors provide a census of proteins belonging to Two Component Systems and Phosphorelays. Additionally they performed a study of its genetic organization (operons/ regulons) and its domain architecture.

Experimental design

Major concerns:
The protocol followed by the authors in "Identification of proteins containing TCS/PR domains" is not clearly presented
Why use Blast, Prosite and HMMer. If HMMer alone is much more sensitive.
Additionally, is not presented which profiles were used to define the different domains analyzed.
It is not clear whether the authors used Pfam profiles, such us:
RR (Response regulator)
Pfam: Response_reg (PF00072)
or
HPt (Histidine-containing Phosphotransfer)
Pfam: Hpt (PF01627)

The analysis of the domain architecture is not very detailed (as it only considers three domains) . Architecture of these proteins is often much more complex, for example with associated domains such as: CheW (PF01584) or P2 (PF07194), among others.

Validity of the findings

I was unable to find novel biological insight in the presented data by Salvado et al. Additionally, published works (cited by the authors) substantially overlap with the analysis presented here.

Reviewer 3 ·

Basic reporting

1. The article language should be improved, a careful proofreading by a native speaker is recommended. Some of the language problems:

Grammatical errors, e.g.:
"associated to", line 21
"domains organize". line 26
"percentage fusion of domains", line 101
"proteins where", line 146, should be "proteins were"
"we might also found", line 183

2. Some factual mistakes/shortcomings are notable:
The phylum "Hyperthermophilic bacteria" does not seem to be a valid phylum, see e.g. Taxonomy database at NCBI.

Line 384, „genes are expressed in a single transcript”. This statement is simply not true. Generally, separate genes have their own, separate transcripts.

The authors repeatedly discuss "coordinated expression" when they really mean proximal location. This may mean correlated expression, but the Authors make an overstatement, because they do not study gene expression.

3. A confusing, unclear language:
In the Abstract, the terms "circuits" and "topologies" may be unclear to a general reader.

In line 248, the Authors state that a model is rejected but do not explain what statistical test is used.

Lines 285-287 contain an extremely confusing language: „HK domains contained motifs similar to those that contain the conserved histidine residue that is needed for HK signal transduction. Hence, these proteins appear to be active HK proteins”. Could the Authors just say if the important His is conserved in their alignments? Just noticing a „similar” motif is not sufficient for any functional conclusions.

4. Too much text is devoted to just repeating the data from long tables.

Experimental design

1. This reviewer is not a statistician, but still the formulae proposed by the authors for probability calculations look suspicious to him.
For example, let's consider Equation 1, and a proteome consisting of three proteins, one of which being an RR protein. Then, P=3 and nRR=1.
Then, according to formula of Eq. 1, probability F(HK-RR) is greater than one. The authors should either correct their formulae or explain more convincingly how they are derived and justified.
Perhaps it would be more straightforward if the authors calculated simply probability of having a consecutive gene pair containing HK and RR domains, and similar probabilities.

2. What were the queries used to screen proteomes for the domains of interest (HK, RR etc)? Just quoting Prosite website is not enough, identifiers of query sequences should be provided.

3. This reviewer believes the Authors were wrong in removing "hypothetical proteins" from the main analysis. Actually, a large proportion of proteomes that are predicted from genomic sequences may end up annotated as "hypothetical proteins", while they may be not more "hypothetical" than proteins annotated otherwise.

The authors were also wrong in removing from the main analysis proteins "annotated as something other than a TCS/PR". This did not allow for possible misannotations. This reviewer thinks that the Authors should probably trust more their own analysis rather than automated annotations.

4. The Authors claim that a quadratic relationship is visible in Fig. 4. This reviewer cannot see this kind of relationship. Actually, the reviewer cannot see any clear relationship in that figure. The red straight line appears to be a very poor fit. By showing the staight line, the Authors contradict their own claim of a quadratic relationship. Moreover, the figure legend does not explain which parts (a and b) relate to eukaryotes and prokaryotes. Moreover, the relatively large R2 quoted for eukaryotes appears to be a result of just two outlier data points.
Even further, the authors do not explain what statistical model is used to derive R2 for data shown in Figure 4.

5. The Authors did not explain if they took any effort to address the many genomes of closely related strains available in sequence databases. If used without any filtering, these genomes and proteome may bias the results by introducing many sets of almost-identical proteins. This risk should be at least addressed in the Discussion.

Validity of the findings

If the reviewer's worries regarding the statistics are addressed, the findings may be useful and meaningful.

Additional comments

None.

---

## Round 0.2 · accepted · Accept

Dear Dr. Alves,

Please address the remaining minor comments the referees have pointed out.

Reviewer 1 ·

Basic reporting

Nothing to add

Experimental design

Description is much better now.

Validity of the findings

Findings in this article are just additions to previous knowledge. This work is fully quantitative, not qualitative.
It's a pity authors did not want to do any other work than pure search of new TCS/PR genes/proteins. It's just a very weak paper.
In my opinion, the most important goal of this work is TO ADD this new data to existing databases. Leaving it as another additional file/website is a waste of energy. The more compact the data is the more scientists can use it.

Additional comments

I very much recommend to add your data to known databases. Otherwise it is another waste of energy.

Reviewer 3 ·

Basic reporting

No Comments

Experimental design

No comments

Validity of the findings

Findings are valid

Additional comments

The changes improved the paper. Inclusion of hypothetical proteins in the analysis is appreciated.

The only change this reviewer would suggest is a grammatical correction in the Title.
Instead of
"phosphorelays proteins"
it should be
"phosphorelay proteins"